# Found in the Middle: How Language Models Use Long Contexts Better via Plug-and-Play Positional Encoding

**Zhenyu Zhang**[1*], **Runjin Chen**[1], **Shiwei Liu**[2], **Zhewei Yao**[3], **Olatunji Ruwase**[3],
**Beidi Chen**[4], **Xiaoxia Wu**[3†], **Zhangyang Wang**[1†]

[1]University of Texas at Austin, [2]University of Oxford, [3]Microsoft, [4]Carnegie Mellon University
[*] Work done during internship at Microsoft, [†] Equal advising

## Abstract

This paper aims to overcome the "lost-in-the-middle" challenge of large language models (LLMs). While recent advancements have successfully enabled LLMs to perform stable language modeling with up to 4 million tokens, the persistent difficulty faced by most LLMs in identifying relevant information situated in the middle of the context has not been adequately tackled. To address this problem, this paper introduces Multi-scale Positional Encoding (Ms-PoE) which is a simple yet effective plug-and-play approach to enhance the capacity of LLMs to handle the relevant information located in the middle of the context, without fine-tuning or introducing any additional overhead. Ms-PoE leverages the position indice rescaling to relieve the long-term decay effect introduced by RoPE, while meticulously assigning distinct scaling ratios to different attention heads to preserve essential knowledge learned during the pre-training step, forming a multi-scale context fusion from short to long distance. Extensive experiments with a wide range of LLMs demonstrate the efficacy of our approach. Notably, Ms-PoE achieves an average accuracy gain of up to 3.8 on the Zero-SCROLLS benchmark over the original LLMs. Code are available at `https://github.com/VITA-Group/Ms-PoE`.

## 1 Introduction

Effective long-sequence reasoning in large language models (LLMs) is crucial for a wide range of applications [1, 2], from understanding extensive texts [3, 4] and managing day-long conversations [5, 6] to code generation [7, 8] and science discoveries [9, 10]. Recent system support advancements [11, 12] have enabled training transformers for any $L$ sequence length even with $O(L^2)$ computational complexity. This is exemplified by models such as MPT [13] and Mistral [14] pre-trained with sequence lengths 16k and 32k respectively.

Nevertheless, emerging research reveals the constrained efficacy of LLMs in managing tasks requiring long contextual understanding. Particularly, [15] demonstrated a substantial degradation in LLMs' performance when crucial information is positioned amidst a lengthy context, a phenomenon they refer to as "lost-in-the-middle". One explanation is about the use of rotary positional embedding (RoPE) [16], a prevalent positional encoding technique used in open-source LLMs. As a relative position embedding, RoPE incorporates a long-term decay property, predisposing the model to prioritize current/nearby tokens while paying less attention to further ones. [17] identified a surprising trend attributed to the Softmax operation where attention scores are disproportionately allocated into initial tokens, irrespective of their relevance to the language modeling task. Despite the presence of considerable redundancy in long-context inputs [18], crucial information may be located across different positions. The inclination of LLMs to overlook the middle section presents a challenge for their applications, particularly in the context of long-context reasoning. Several approaches successfully extend pre-trained LLMs with context up to extreme token length, either through sparse

selection of crucial tokens during generation [17, 18, 19] or by modifying positional encoding [20, 21]. *Nevertheless, these approaches primarily aim to extend the context length of LLMs and, consequently, fall short in addressing the "lost-in-the-middle" problem when applied out-of-the-box.*

Efforts have been made to enhance LLMs' capacity to capture vital information located within the middle of the context. These include extra memory bank [22], reordering the input context based on relevance [23, 24], enhancing the information searching and reflection ability via attention strengthening tasks [25, 26], splitting the input into short segments and applying short-text models [27]. For example, [23] empirically discovered that LLMs tend to emphasize more on the current window while still paying more attention to the relevant text than distracting content. They subsequently introduced "attention sorting" where the main idea is iteratively sorting documents based on their attention scores, such that critical information will likely be placed at the end, to fit the position-biased nature of RoPE. [24] conducted parallel runs of LLMs with different RoPE angles, thereby mitigating the risk of overlooking crucial information

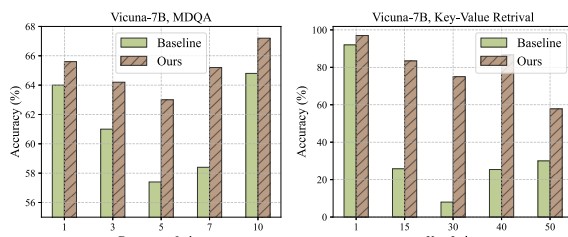

Figure 1: The x-axis illustrates the placement of essential information within the prompt, ranging from start to end. The green bar serves as a standard baseline, illustrating the "lost-in-the-middle" phenomenon. We introduce our method, **M**ulti-**s**cale **Po**sition **E**ncoding (Ms-PoE), which requires neither additional fine-tuning nor increased memory usage. Instead, it involves a simple remapping of the position embedding depicted in Figure 2, which enables the important information in the middle to be detected effectively (brown bars). For more details, see Section 4.2 and Figure 5.

through a weighted sum of the outputs. These approaches usually require additional memory or multiple inference runs, which can be expensive for LLMs.

In this paper, we aim to address the "lost-in-the-middle" problem by reintroducing the concept of multi-scale features from computer vision into the context of Transformer-based LLMs. Multi-scale features, well-established in Inception-style models [28, 29, 30], utilize parallel employment of kernels with different sizes to fuse multi-scale information, spanning short to long distances. Introducing multi-scale operations into LLMs intuitively can help compensate for crucial information located in the middle, which might be easily overlooked by full attention operation. Unlike modifying the attention module to form multi-scale attention, we choose to re-scale the indices of positional encoding. This decision is grounded not only in its effectiveness in easily adjusting the scale of the context window by simply changing the position indices [20] but also in the potential of down-scaling the position indices to relieve the long-term decay property introduced by RoPE. However, this approach was initially introduced to extend context windows, and its performance regarding the "lost-in-the-middle" problem remains uncertain for several reasons: (i) Indice re-scaling forces position embeddings of original context window to reside in a narrower region, leading to performance degradation in the original context window as shown in [20]. (ii) Uniformly applying the same scaling ratio throughout the entire model might be sub-optimal to preserve essential knowledge learned during pre-training; (ii) Fine-tuning is necessary for the original approach, albeit minimal. The impact without fine-tuning remains unknown.

To this end, we systematically visit the position indices scaling regarding the "lost-in-the-middle" problem and counter-intuitively discover that it is possible to slightly mitigate the "lost-in-the-middle" issue if we carefully choose the scaling ratio to be around 1.5-2. Additionally, we observe that different attention heads exhibit varying sensitivity to the position shift of the relevant document. Some attention heads are "position-aware", consistently capturing relevant information even with position shifts, while others may occasionally capture position changes, and some heads are completely insensitive to position changes. This highlights the need to treat attention heads separately when re-scaling position indices.

**Contribution.** Inspired by the above observations, we introduce Multi-scale Positional Encoding (Ms-PoE), a simple yet effective plug-and-play approach that can enhance the long-context reasoning capability of pre-trained LLMs without requiring fine-tuning or introducing any additional overhead. Ms-PoE meticulously assigns distinct scaling ratios to different attention heads, with the scaling factor monotonically increasing from "position-aware" heads to "position-unaware" heads. This enables

us to improve long-context ability by re-scaling position indices to shorter values while preserving essential knowledge acquired during the pre-training phase. The efficacy of Ms-PoE is substantiated through extensive experiments. By simply re-scaling the indices of positional encoding, Ms-PoE consistently enhances the performance of various LLMs including Llama-2 [31], StableBeluga [32] and Vicuna [33] on the ZeroSCROLLS [34], achieving a notable accuracy gain of up to 3.8.

## 2 Background and Related Works

In this section, we provide a concise overview of the background knowledge and recent literature about the generative inference process of Large Language Models (LLMs), their abilities for long-context reasoning, and details of positional encoding.

### 2.1 Generative Inference of LLMs

The generative inference process in LLMs can be categorized into two distinct phases: ① Prefilling Stage: In this initial phase, LLMs receive an input sequence containing detailed instructions that define a specific generation goal. Throughout this stage, intermediate Key and Value embeddings are generated at each layer and stored in memory, commonly referred to as the KV cache. ② Decoding Stage: This phase involves retrieving embeddings from the KV cache to generate new tokens. The decoding process is inherently iterative, where each newly generated token serves as input for the subsequent token generation. In real-world LLM deployment, the cumulative length of input sequences and the subsequently generated text can reach several thousand or even millions of tokens, presenting significant challenges for the LLMs' long-context reasoning capability.

### 2.2 Long Context Reasoning

Two challenges for LLMs in handling long-context reasoning tasks. One is to extend the context window to process sentences that exceed the pre-trained window length. Another is the "lost-in-the-window" issue where LLMs likely overlook the information located in the middle of the sentences.

The reason for the former challenge is that open-source LLMs are usually pre-trained with fixed sequence lengths, such as 4096 for Llama-2 [31]. When the sequence length surpasses the predefined context length used in pre-training, LLMs often suffer from performance collapses and thus generate incoherent or fragmented text. Recent efforts to address this issue can be broadly categorized into two streams. Recently, several works have been proposed to address this issue, which can be broadly categorized into two streams. The first one explores from the expansion of positional encoding, with notable contributions including PI [20], CLEX [35], YaRN [36], Self-Extend [21]. On the other hand, some works modify the attention mechanism, such as StreamingLLM [17], LM-Inifinite [19], $H_2O$ [18], TOVA [37], Zebra [38], and Activation Beacon [39]. These approaches have successfully expanded the contextual window with minimal or no additional training overhead.

Despite the extended context window, LLMs still face a significant challenge in long-context inference due to the uneven utilization of lengthy inputs. [15] conducted a pivotal investigation, revealing that LLMs tend to overlook the middle portion of the input. This bias compromises the practical application of LLMs, as critical information may be located in the middle part of the input, leading to unreliable outputs. To tackle this issue, [23] introduced 'attention sorting' to reorder inputs, placing critical information at the end. However, this method's reliance on potentially biased attention scores to identify crucial content may compromise its reliability, and the prerequisite knowledge of document count in inputs may affect its effectiveness. [24] utilize Attention Buckets, an ensemble approach that combines multiple forward processes with positional modifications. However, this technique necessitates a considerably higher computational cost. Other general approaches for enhancing long-context reasoning include prompt compression [40], retrieval augmentation [26], and inference refinement by constructing memory trees [41] while these approaches typically necessitate extra LLMs' assistance or bring extra computational cost.

### 2.3 Positional Encoding

For effective processing of long contexts, LLMs necessitate the explicit encoding of positional information. Common techniques include absolute positional embedding and relative positional encoding. Absolute positional embedding integrates word embeddings with an additional positional

vector based on the token's absolute position, which can be either fixed [42] or learnable [43, 44, 45, 46, 47]. In contrast, relative positional encoding, increasingly popular in contemporary LLMs, encodes the relative distances between tokens instead of their absolute positions. Notable among these are Rotary Position Embedding (RoPE) [16] that widely implemented in models like Llama [31], Falcon [48], Mistral [49], and ALiBi [50], which used in MPT [13].

**RoPE.** The primary goal of RoPE [16] is to encode positional information such that the inner product of the query and key embeddings inherently contains the relative position information:

$$f(\mathbf{q}_m, m)^T f(\mathbf{k}_n, n) = g(\mathbf{q}_m, \mathbf{k}_n, m - n)$$

Here, $f$ is the positional encoding function applied to the query and key embeddings at positions $m$ and $n$, respectively. To satisfy this condition, the function $f$ is defined as a vector-valued complex function, as follows:

$$f(\mathbf{x}, m) = \mathbf{x}e^{im\theta}$$
$$= [(x_1 + ix_2)e^{im\theta_1}, (x_3 + ix_4)e^{im\theta_2},$$
$$..., (x_{l-1} + ix_l)e^{im\theta_{l/2}}]^T$$

In this equation, $l$ represents the dimension of the embeddings, $\theta_k = 10000^{-2k/l}$, and $i$ is the imaginary unit. For calculating the attention score, RoPE considers the real part of the product, specifically $\mathrm{Re}(f(\mathbf{q}_m, m)^T f(\mathbf{k}_n, n))$. This approach allows RoPE to effectively integrate relative positional information into the attention mechanism of transformer models.

## 3 Methodology

In this section, we present the details of our Multi-Scale Positional Encoding (Ms-PoE) approach. Section 3.1 demonstrates that the context utilization of LLMs can be directly enhanced by re-scaling the positional information without incurring extra training costs. Then, Section 3.2 analyzes the properties of various attention heads in LLMs. Section 3.3 outlines the detailed pipeline of Ms-PoE.

### 3.1 Positional Re-scaling Improves Context Utilization

Current LLMs tend to neglect information located in the middle of the context, despite its potential relevance. This "lost in the middle" phenomenon likely arises from two contributing factors: (i) Casual Attention, where preceding tokens undergo a higher number of attention processes, leading LLMs to disproportionately favor initial tokens. This phenomenon has been demonstrated in recent research which highlights the pivotal role of the initial tokens in model generation [19, 17], with these starting tokens consistently accumulating higher attention scores [18]. (ii) The utilization of RoPE [16] introduces a long-term decay effect, diminishing the attention score of distantly positioned yet semantically meaningful tokens. The combination of these factors contributes to LLMs neglecting the context in the middle part. To tackle this issue and improve the context utilization of LLMs, a seemingly unreasonable yet remarkably effective strategy is to down-scale positional information [38]. Formally, RoPE encodes the position as

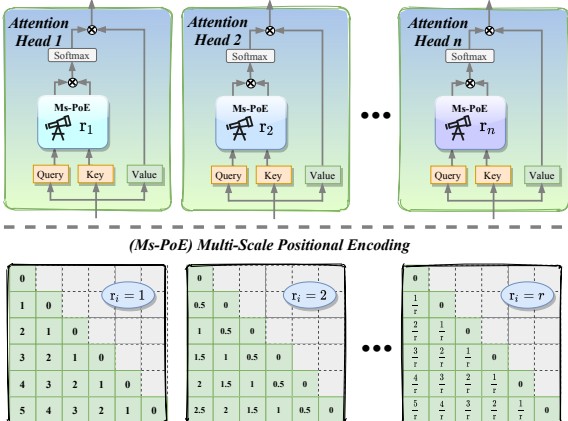

Figure 2: Illustration of our Multi-scale Positional Encoding (Ms-PoE) framework. The top figure demonstrates the implementation of Ms-PoE with various scaling ratios in different attention heads, marked with different colors. The bottom figure shows the position details of each head, in which the first matrix ($r_i = 1$) represents the original RoPE.

$f(\mathbf{x}, m) = \mathbf{x}e^{im\theta}$. By substituting the position $m$ with $\frac{m}{r}$, we can force the long-distance tokens to

reside in the shorted range, which can potentially alleviate the long-term decay effects by a factor of $r$. In the following sections, we conduct experiments to evaluate how LLMs' context utilization responds to varying re-scaling ratios $r$.

**Details.** Experiments are conducted using Llama-2-7B-Chat [31] and Vicuna-7B [33] on the Multi-Document Question Answering (MDQA) task [15]. Each question includes ten documents, with only one relevant to the question. By varying the position of the relevant document, we can evaluate LLMs' context utilization properties. For each position of the key document, we calculate the accuracy over 500 samples. And results show in Figure 3 include both the **Average** accuracy over the 10 documents as well as **Gap** accuracy, *i.e.*, the difference between the best and worst accuracy when varying the positions of the relevant document.

**Results.** Figure 3 demonstrates that the gap accuracy can be alleviated via appropriate positional re-scaling. Particularly, we see that the Gap between the best and the worst accuracy is greatly reduced when increasing the re-scaling ratio. An enhanced average accuracy can be observed with a scaling ratio equals near 1.5. Additionally, changing the scaling ratio also affects the favored zone of LLMs. With a small scaling ratio (e.g., 0.5), LLMs tend to focus more on the most recent part of the context, while with a large ratio (e.g., 2.5), LLMs favour the beginning part.

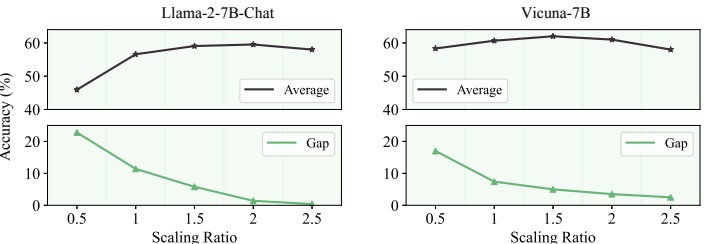

Figure 3: Results of the relationship between positional re-scaling and context utilization. The upper curve illustrates the average accuracy when placing the key document in various positions. The bottom curve indicates the gap between the best and worst accuracy.

**Improving context reasoning via positional re-scaling.** Building upon this, we introduce a plug-and-play treatment for RoPE by re-scaling the position of each token. This approach seamlessly enhances the context utilization of LLMs without requiring additional training or inference overhead. However, there is a trade-off in terms of LLMs favoring certain context regions. For instance, when $r = 0.5$, LLMs achieve peak accuracy when the relevant document is located at the end of the input, while at the beginning for $r = 1.5$. It remains challenging to decide which re-scaling ratio to use, given that we lack prior knowledge of the location of relevant information in real-world applications. Moreover, as the re-scaling ratio increases, LLMs may face the positional out-of-distribution (O.O.D) issue [21, 20], where many position values do not directly exist during pretraining (e.g., using $0.1, 0.2, ..., 0.9$ for position when LLMs only recognize $1, 2, ..., 9$ during pretraining), potentially reducing their average reasoning ability. To tackle these challenges, we investigate the head-wise properties of LLMs and propose a multi-scale positional encoding approach.

### 3.2 Position-Aware Head-Wise Re-scaling Ratio

Inspired by recent works that leverage attention patterns to identify most crucial tokens and optimize inference efficiency [37, 18, 51], we carry out a preliminary study to investigate the interaction between attention patterns and token positions.

**Details.** We visualize the attention patterns of the most recent query with results collected from Vicuna-7B on the MDQA task, following [37]. In the same input sample, we manually switch the position of the relevant document from the beginning to the end and illustrate the attention scores across different positions.

**Observation.** We observe the presence of "position-aware" attention heads capable of capturing relevant information even when its position is shifted. As an example, we select the eighth attention head in the fifteenth layer, depicted in the bottom of Figure 4, while consistent observations can be drawn across different layers and input samples. Firstly, most attention scores are near zero and can be ignored, consistent with other studies highlighting high sparsity in attention blocks [18, 52, 53]. For the remaining positions, these "position-aware" attention heads can capture important information across positions, with attention patterns shifting as the position of relevant tokens changes. However,

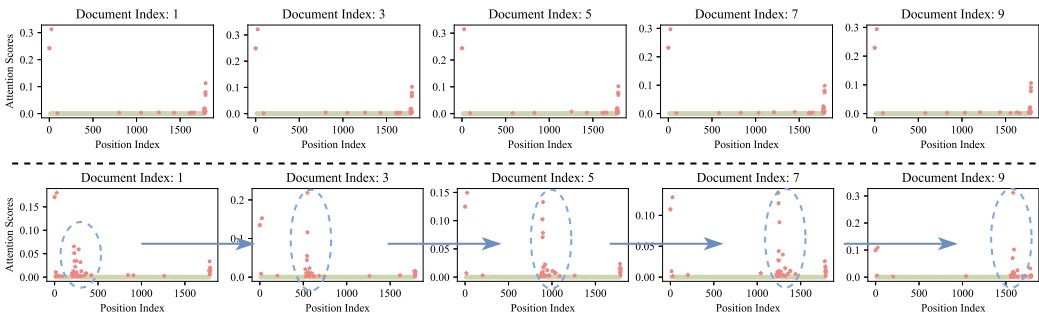

Figure 4: Visualization of attention pattern of the most recent query within two different attention heads. **Top:** Results of the 12th attention head in the 15th layer. **Bottom:** Results of the 8th attention head in the 15th layer. The most recent query remains unchanged while varying the position of the crucial document. More examples are reported in Figure 6 in the appendix.

for other attention heads (upper subfigure in Figure 4), they fail to capture relevant tokens and only attend to the beginning and end words, contributing to the "lost-in-the-middle" issue.

Based on this observation, we devise a position-aware strategy to adaptively determine the re-scaling ratio via the inherent properties of different attention heads. For the "position-aware" attention heads, we assign a re-scaling ratio close to one to avoid changing their functionality significantly, as altering them too much could degrade performance due to the positional O.O.D issue. On the other heads, we condense their position indices to a higher degree, providing more opportunity to alleviate the persistent bias toward the beginning and recent tokens. To identify the properties of $n_h$ attention heads, we introduce a Position-Awareness Score $\mathcal{S}_P \in \mathbf{R}^{n_h}$ formulated as:

$$\mathcal{S}_P = \frac{1}{l} \sum_{i=1}^{l} (A_i \geq \alpha \frac{1}{l} \sum_{i=1}^{l} A_i) \tag{1}$$

In Equation 1, $A$ represents the attention score vector of the most recent query, and $\alpha$ is a hyper-parameter determining the threshold of effective attention scores. In all experiments, we default to using $\alpha = 3$, and the corresponding important tokens are highlighted in Figure 4, which are shown in red. In the spirit of numerous studies that investigate the outlier properties in LLMs [17, 54, 55], we utilize $\mathcal{S}_P$ to evaluate the ratio of effective attention tokens, where a larger $\mathcal{S}_P$ value implies better positional awareness.

### 3.3 Inference with Multi-Scale Positional Encoding

The pipeline for utilizing Multi-Scale Positional Encoding (Ms-PoE) in LLM inference is: Given a pre-trained LLM, we initially replace the original rotary positional encoding with Ms-PoE. As illustrated in Figure 2, Ms-PoE condenses the positional indices of RoPE and employs different re-scaling ratios for each attention head. The re-scaling ratios are assigned during the prefilling stage, where we first calculate the distribution of attention scores for the most recent query and obtain the corresponding position-awareness score for each attention head. Larger re-scaling ratios are subsequently allocated to attention heads exhibiting smaller position-awareness scores. And the set of re-scaling ratios **r** defaults to a **linear** range from $R_{min}$ to $R_{max}$. For example, the $i$th sorted-head would be using re-scaling ratio

$$r_i = R_{min} + (i-1)(R_{max} - R_{min})/(n_h - 1) \tag{2}$$

Once the re-scaling ratios are assigned, they remain fixed in the subsequent decoding stage. We consistently using $R_{min} = 1.2$ and $R_{max} = 1.8$ is our experiments.

## 4  Experiments

The goal of this section is to demonstrate Ms-PoE, a plug-and-play positional encoding capable of enhancing the context utilization of LLMs, and consequently improving the quality of generation across diverse models and downstream reasoning tasks. Our main results can be summarized below.

Table 1: Comparsion results on ZeroSCROLLS [34] benchmarks. The evaluation metrics for various tasks are tailored as follows: GovReport, SummScreenFD, QMSum, and SQuALITY utilize the geometric mean of Rouge-1/2/L scores. Qasper and NarrativeQA are assessed through the F1 score, while BookSumSort employs the concordance index.

| Models | Methods | GovReport | SummScreenFD | QMSum | SQuALITY | Qasper | NarrativeQA | BookSumSort | Average |
|---|---|---|---|---|---|---|---|---|---|
| Llama-2-7B-Chat | Baseline | 16.8 | 14.1 | 15.2 | 19.5 | 21.9 | 14.4 | 3.1 | 15.0 |
| Llama-2-7B-Chat | Ours | 17.7 (+0.9) | 14.2 (+0.1) | 15.8 (+0.6) | 19.9 (+0.4) | 25.1 (+3.2) | 17.7 (+3.3) | 5.8 (+2.7) | 16.6 (+1.6) |
| Llama-2-13B-Chat | Baseline | 15.4 | 12.3 | 15.1 | 18.9 | 19.0 | 15.0 | 5.7 | 14.5 |
| Llama-2-13B-Chat | Ours | 16.5 (+1.1) | 13.1 (+0.8) | 15.5 (+0.4) | 19.2 (+0.3) | 20.8 (+1.8) | 17.0 (+2.0) | 5.9 (+0.2) | 15.4 (+0.9) |
| StableBeluga-7B | Baseline | 14.9 | 13.8 | 14.7 | 17.9 | 28.1 | 16.8 | 9.2 | 16.5 |
| StableBeluga-7B | Ours | 16.6 (+1.7) | 14.2 (+0.4) | 15.2 (+0.5) | 18.7 (+0.8) | 36.9 (+8.8) | 18.0 (+1.2) | 14.2 (+5.0) | 19.1 (+2.6) |
| StableBeluga-13B | Baseline | 5.7 | 7.1 | 12.9 | 13.3 | 19.2 | 13.4 | 4.8 | 10.9 |
| StableBeluga-13B | Ours | 7.4 (+1.7) | 7.4 (+0.3) | 12.8 (-0.1) | 13.2 (-0.1) | 20.8 (+1.6) | 13.4 (+0) | 5.6 (+0.8) | 11.5 (+0.6) |
| Vicuna-7B | Baseline | 16.2 | 13.7 | 15.1 | 18.9 | 24.3 | 13.7 | 3.3 | 15.0 |
| Vicuna-7B | Ours | 20.2 (+4.0) | 14.5 (+1.8) | 15.4 (+0.3) | 19.8 (+0.9) | 34.7 (+13.4) | 16.2 (+2.5) | 10.5 (+7.2) | 18.8 (+3.8) |
| Vicuna-7B-16K | Baseline | 20.2 | 13.9 | 16.2 | 20.1 | 32.3 | 18.8 | 29.9 | 21.6 |
| Vicuna-7B-16K | Ours | 21.4 (+1.2) | 14.3 (+0.4) | 16.2 (+0) | 20.2 (+0.1) | 37.8 (+5.5) | 21.0 (+2.2) | 43.3 (+13.4) | 24.9 (+3.3) |

In Section 4.1, we demonstrate that Ms-PoE consistently enhances reasoning over long contexts for a range of tasks in the ZeroSCROLLS benchmarks [34], all without the need for additional training. Additionally, Ms-PoE exhibits superior performance when compared to other methods in the field, including PI [20] and Self-Extend [21]. Detailed comparison results are shown in Tables 1 and 2.

In section 4.2, we highlight that Ms-PoE improves the context utilization and achieves consistent improvement when varying the position of critical information, as shown in Figure 1 & 5.

In Section 4.3, we conduct multiple ablation studies to assess the effectiveness of Ms-PoE under different scaling ratios and selection strategies. Results are reported in Table 3 & 4.

## 4.1 Enhanced Generation Quality

We empirically validate the ability of Ms-PoE to enhance long-context reasoning with a noteworthy improvement up to $13.4$ without additional training overhead. Notably, our approach surpasses other competitive baselines, demonstrating improvements from $2.64$ to $43.72$.

**Experimental Setup.** In our experiments, we select seven representative LLMs, including Llama-2-chat-7B and 13B [31], StableBeluga-7B and 13B [32], and Vicuna-7B [33], along with its longer-context version (Vicuna-7B-16K). To comprehensively evaluate the long-context reasoning abilities of LLMs, we choose seven tasks from ZeroSCROLLS [34], spanning all four task categories: ① Document Summarization (Government and SummScreenFD), ② Query-Based Summarization (QMSum and SQuALITY), ③ Question Answering (Qasper and NarrativeQA), and ④ Information Aggregation (BookSumSort). We also compare Ms-PoE with other competitive methods on additional generation tasks, including Multi-document Question Answering (MDQA) and Key-Value Retrieval [15].

**Main Results.** Table 1 summarizes the main results, yielding several key observations: (i) By simply substituting the original positional encoding module with our Ms-PoE, the performance of LLMs consistently improves across all tasks without additional training, resulting in an average performance enhancement ranging from **0.6** to **3.8**; (ii) These improvements hold consistently across different model sizes of 7 billion and 13 billion parameters; (iii) The efficacy extends to LLMs with varying sequence lengths, such as Vicuna-7B and its extended version, Vicuna-7B-16K, both showing improvements from **3.3** to **3.8**.

**Outperform other competitive methods.** We conduct a thorough comparison between Ms-PoE and other competitive methods, including Positional Interpolation (PI) [20] and Self-Extend [21], both of which modify position indices without utilizing head-wise properties. For PI, we employ the scaling ratio as the average value of our method while for Self-Extend, we set the group size as 2 with the local window size as $1024$. The results presented in Table 2 consistently showcase the superiority of our approach over other baselines, demonstrating improvements of up to $3.92$ and $43.72$ for MDQA and Key-Value Retrieval, respectively. Such improvements might come from two primary factors. Firstly, the incorporation of head-wise properties offers a more adaptive strategy for positional modification. Secondly, our approach enhances the general context utilization ability. Notably, our approach demonstrates superiority even when the core document or key is positioned at the end of the input, surpassing other baselines with improvements ranging from $2.4$ to $27.8$.

This performance surpasses the recent work [23], which addresses the "lost-in-the-middle" effect by reordering key documents and placing them at the end of the input. When the identified core document is already located at the recent area, such method can not gain further improvements, while our approach offers a *fine-grained* strategy to improve context utilization.

Table 2: Comparsion results with other competitive methods on MDQA and Key-Value Retrival. Results are reported in accuracy.

| Models | Methods | MDQA | | | | | |
| | | 1 | 3 | 5 | 7 | 10 | Average |
|---|---|---|---|---|---|---|---|
| | Baseline | 64.0 | 61.0 | 57.4 | 58.4 | 64.8 | 61.12 |
| | PI | 65.2 | 62.4 | 60.0 | 60.4 | 64.0 | 62.40 |
| Vicuna-7B | Self-Extend | 64.7 | 63.7 | 61.4 | 59.8 | 62.0 | 62.32 |
| | Ms-PoE | **65.6** | **64.2** | **63.0** | **65.2** | **67.2** | **65.04** |
| Models | Methods | Key-Value Retrieval | | | | | |
| | | 1 | 15 | 30 | 40 | 50 | Average |
| | Baseline | 92.0 | 25.8 | 8.0 | 25.4 | 30.0 | 36.24 |
| | PI | 96.4 | 76.4 | 61.4 | 64.6 | **57.8** | 67.60 |
| Vicuna-7B | Self-Extend | 88.6 | 63.8 | 76.2 | 59.4 | 42.0 | 66.00 |
| | Ms-PoE | **97.0** | **83.4** | **75.0** | **86.6** | 57.8 | **79.96** |

## 4.2 Superior Context Utilization

We assess the context utilization ability of our approaches on two tasks, including multi-document question answering (MDQA) and key-value retrieval (KV retrieval) tasks from [15]. Such tasks provide a good input structure and offers the flexibility to switch the position of crucial information, thus evaluate the context utilization ability of LLMs.

**Experimental Setup.** In the MDQA task, each input sample comprises ten documents and one question, with only one document being relevant to the question. For the KV retrieval tasks, there are 50 key-value pairs with one question querying the value of the chosen key. In both tasks, we systematically switch the important document or key-value pair

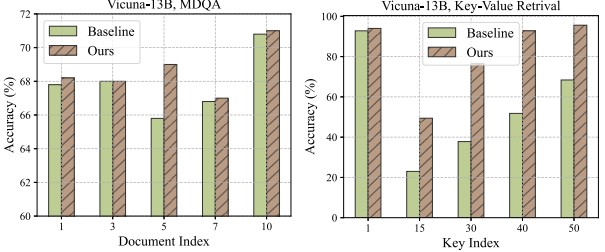

Figure 5: Comparison results for the multi-document question answering (MDQA) and key-value retrieval (KV retrieval) tasks. Each subfigure depicts the comparison when varying the position of critical information from the beginning to the end. For Vicuna-7B, please refer to Figure 1.

from the beginning to the end and report the accuracy of the generated context. All results are averaged across 500 samples. The **Gap** accuracy metric is employed to assess the context utilization ability of LLMs, defined as the gap between the best and worst accuracy when varying the position of important information.

**Main Results.** As depicted in Figure 5 and 1, Ms-PoE demonstrates consistent improvement across different models, tasks and critical positions. Even when the important information exists in the sweet region (beginning and end) of the input, Ms-PoE achieves significant performance improvements ranging from 3% to 6%, highlighting its efficacy in enhancing generation quality. Moreover, the "lost-in-the-middle" issue is notably alleviated, with Ms-PoE quantitatively reducing the gap accuracy by approximately 2% to 4%, showcasing improved context utilization.

## 4.3 Ablation Study and More Investigation

This section conducts a further evaluation of the effectiveness of Ms-PoE by addressing the following questions: *Q1:* How does the effectiveness of Ms-PoE relate to the head-wise selection strategy of the scaling ratio? *Q2:* How does the model perform with different scaling ratios?

*A1:* **Positional awareness metrics achieve superior performance compared to other strategies.** For a set of scaling ratios $\mathbf{r} \in R^{n_h}$, where $n_h$ is the number of attention heads, and using scaling ratios linearly ranging from 1.2 to 1.8, we evaluate various strategies for assigning these ratios to different attention heads. These strategies include: ① `Random`, which randomly assigns the scaling ratios to each head within each layer;

Table 3: Ablation results of different ordering metrics. Experiments are conducted on Multi-Documents Question Answering task with the Vicuna-7B model.

| Methods | Begin | Middle | End | Average |
|---|---|---|---|---|
| Baseline | 64.0 | 57.4 | 64.8 | 62.1 |
| Random | 64.5 | 55.0 | 65.5 | 61.7 |
| Sequential | 60.5 | 54.5 | 58.5 | 57.8 |
| Entropy | 63.5 | 59.5 | 64.0 | 62.3 |
| Position-Awareness | **65.6** | **63.0** | **67.2** | **65.3** |

② `Sequential`, performing the assignment based on the original head order; ③ `Entropy`, where we follow metrics measuring the sparsity level of attention scores [56]. Larger entropy implies less sparse attention scores, indicating the model attends to more tokens rather than just the beginning and end words, so we assign a scaling ratio near to 1, and vice versa for larger ratios. Results in Table 3 demonstrate that the proposed position-awareness effectively captures the head-wise properties of LLMs, enhancing performance when critical information is located at various positions—beginning, middle, or end. This leads to an average accuracy gain of 3.2 (65.3 v.s. 62.1).

*A2:* **Ablation study of the scaling ratios.** We first examined the effect of uniform scaling ratios across all heads on model performance. Our findings, outlined in Table 4, indicate that adjusting the scaling ratio between 0.5 and 2.5 can significantly enhance generative performance and mitigate the "lost-in-the-middle" effect by 1.0% (63.1% *v.s.* 62.1%), particularly with a ratio of 1.5. Further testing with an average ratio of 1.5 across all heads revealed that an optimal range exists between 1.2 and 1.8, leading to an additional 2.2% (65.3% *v.s.* 63.1%) accuracy improvement with our approach, Ms-PoE. Based on these results, we established these ratios as our experimental standard.

Table 4: Ablation results of the condensing ratios. Experiments are conducted on Multi-Documents Question Answering task with the Vicuna-7B model.

| Scaling Ratio | Begin | Middle | End | Average |
|---|---|---|---|---|
| 1 | 64.0 | 57.4 | 64.8 | 62.1 |
| 0.5 | 56.0 | 51.0 | 68.0 | 58.3 |
| 1.5 | 65.2 | 60.0 | 64.0 | 63.1 |
| 2 | 61.5 | 59.0 | 62.5 | 61.0 |
| 2.5 | 59.5 | 57.5 | 57.0 | 58.0 |
| $0.8 \rightarrow 2.2$ | 53.5 | 59.5 | 67.5 | 60.2 |
| $1 \rightarrow 2$ | 61.0 | 57.0 | 63.0 | 60.3 |
| $1.2 \rightarrow 1.8$ | **65.6** | **63.0** | **67.2** | **65.3** |
| $1.4 \rightarrow 1.6$ | 65.5 | 59.0 | 63.0 | 62.5 |

## 5 Conclusion

In this paper, we present a plug-and-play strategy designed to address the "lost-in-the-middle" challenge observed in LLMs. This challenge stems from the persistent bias exhibited by LLMs towards the beginning and local content within the input, leading to the neglect of crucial information in the middle. Our investigation reveals the effects of position indice rescaling and the head-wise position-awareness property, leading to the introduction of Multi-scale Positional Encoding (Ms-PoE). This approach enhances the capability of LLMs to effectively capture information in the middle of the context without the need for additional fine-tuning. Comprehensive experiments conducted on Zero-SCROLLS benchmarks, multi-document question-answering tasks, and key-value retrieval tasks confirm the effectiveness of Ms-PoE.

## 6 Acknowledgements

We thank Dr. Yuandong Tian for interesting discussions on this work. Z. Zhang and Z. Wang were in part supported by an Intel Gift Funding.

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

# A More Experiment Results

## A.1 Position-Aware Attention Heads

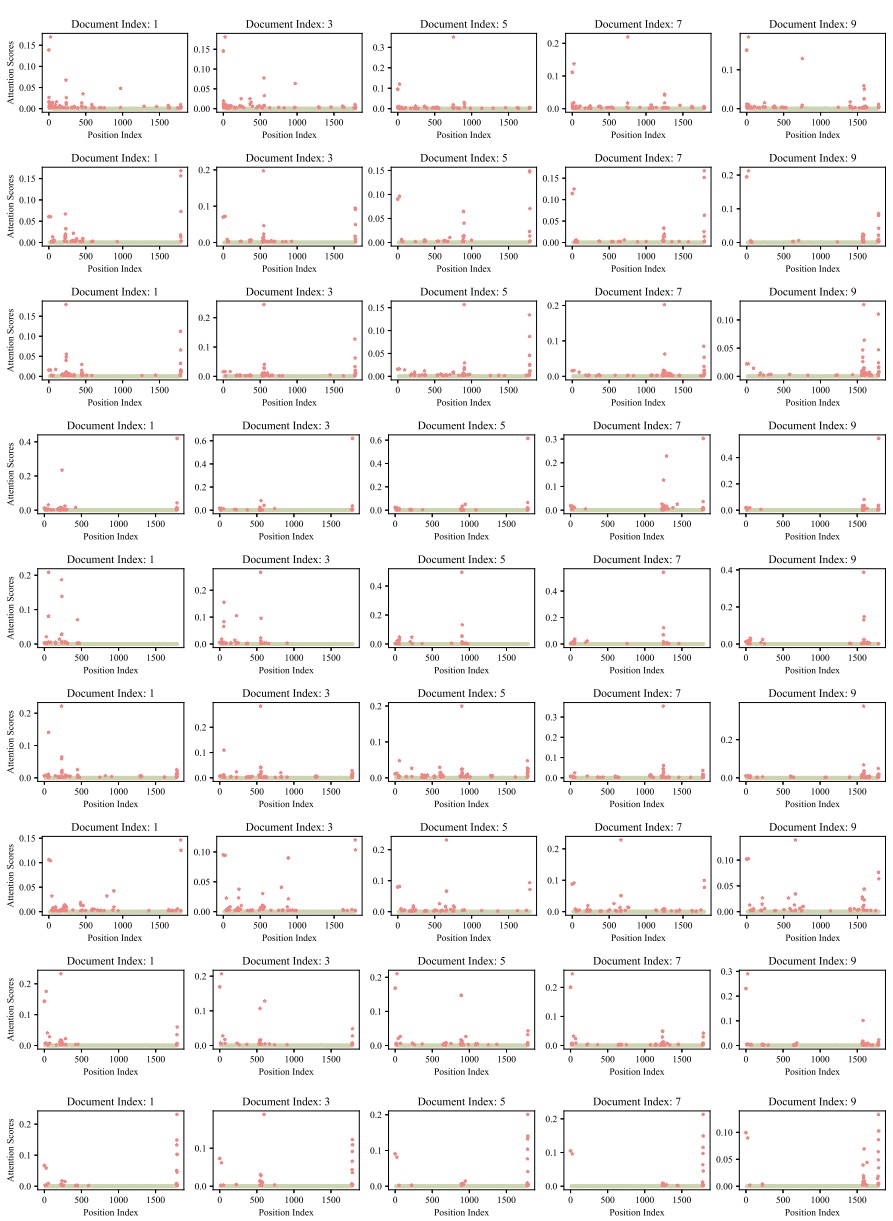

Figure 6: Visualization of "position-aware" attention heads. Each row contains the attention pattern for the same heads when varying the key documents within the inputs.

Figure 6 illustrates the attention patterns of "position-aware" heads. Each row represents the attention pattern of the same head. As the key document is positioned from the beginning to the end, the attention peak gradually shifts, indicating robust positional awareness. It's important to note that we randomly selected 9 attention heads with these "position-aware" properties, and these results were validated with different input samples and layers.

Table 5: Comparison results of Ms-PoE on LongBench-EN benchmark with Llama-2-7B-Chat.

| Methods | MultiFieldQA-en | LCC | GovReport | HotpotQA | Passage Count | Qasper | MultiNews | SAMSum | TriviaQA | PassageRetrieval-en | RepoBench-P | TREC | 2WikiMQA | Average |
|---------|-----------------|-----|-----------|----------|---------------|--------|-----------|--------|----------|---------------------|-------------|------|----------|---------|
| Baseline | 33.51 | 59.77 | 27.97 | 30.10 | 3.74 | 19.27 | 24.36 | 39.45 | 82.81 | 10.00 | 49.22 | 57.33 | 28.14 | 35.82 |
| Ours | 37.33 | 62.03 | 29.87 | 34.08 | 4.60 | 20.96 | 24.69 | 39.79 | 85.28 | 16.67 | 50.11 | 58.67 | 30.19 | 38.02 |

## A.2 Results on LongBench-EN Benchmark

We further evaluate Ms-PoE on the LongBench-EN benchmark [57] that contains 13 tasks aims for long context understanding. Results are reported in Table 5. We can observe that Ms-PoE achieves consistent performance improvement without any finetuning.

