# OpenReview forum: "Found in the Middle: How Language Models Use Long Contexts Better via Plug-and-Play Positional Encoding"
_NeurIPS.cc/2024/Conference — NeurIPS 2024 poster_

### Official Review · Reviewer_LVQT · 2024-06-25

**Soundness:** 3
**Presentation:** 3
**Contribution:** 3
**Rating:** 6
**Confidence:** 4

**Summary:**

Paper addresses the lost-in-the-middle effect, observed in the past for some LLMs, with a method called Multi-scale Positional Encoding (Ms-PoE) where position is encoded using different scales for each attention head. More precisely, for each head a re-scaling ratio is substituting the position m of a token with m/r (r can be 1.5 for instance) and different values of r are used for each attention head. This multi-scale view is evaluated experimentally and authors claim that this mitigates the ‘lost-in-the-middle’ effect.

**Strengths:**

-a new approach for positional encoding Multi-scale Positional Encoding (Ms-PoE)
-approach is plug-and-play (no additional fine-tuning needed)

**Weaknesses:**

-The "lost-in-the-middle" effect is often assumed to impact all LLMs. But  recent research indicates that this effect does not uniformly affect all LLMs, though the effect was observed on models such as Vicuna7B.
	see for instance https://arxiv.org/abs/2403.20262  which shows that not all LLMs display the lost-in-the-middle effect on a long context benchmark

-For evaluation,  the ZeroSCROLLS benchmark was used but authors could have considered  incorporating other benchmarks designed for long context scenarios, such as "needle in the haystack" tasks. There are also many other relevant benchmarks available, including: NarrativeQA; LongEval; LongBench; LongBench-Chat; Loogle;  ∞Bench and Long Range Arena (LRA) among others
      => on that aspect paper could have been stronger experimentally if evaluating the approach on more long context benchmarks

**Questions:**

-eq(2): how are Rmin & Rmax fixed ?

-fig 5: why only vicuna model evaluated, does Llama-2-7B-Chat also display a lost-in-the-middle effect ?

-tab.3 are the differences between begin/middle/end results really significant ? A statistical test (for instance one-tailed Welch’s t-test) would have been welcome here

**Limitations:**

No 'limitations' section was provided

---

> ### Author Rebuttal · Authors · 2024-08-07
>
> We sincerely appreciate Reviewer LVQT for supporting our work and providing constructive suggestions. To address Reviewer LVQT’s concerns, we provide point-wise responses below.
>
>
> **[Q1: No Lost-in-the-Middle Effect]** Thank you for the insightful question. We have observed similar findings to those in [1], particularly with **LLaMA-2-7B-Chat**, which does not exhibit the lost-in-the-middle phenomenon but rather shows performance loss at the beginning. The results are reported in Table R1. Additionally, after applying Ms-PoE, consistent improvements are achieved by an average of 1.5 accuracy. This further demonstrates the effectiveness of our approach for enhancing the context utilization of LLM inference. We’ve added the results in the updated draft.
>
> [1] ELITR-Bench: A Meeting Assistant Benchmark for Long-Context Language Models.
>
> Table R1. Comparison results of LLaMA-2-7B-Chat with and without Ms-PoE on the MDQA tasks.
>
> |Key-Document-Index|Beginning|Middle|End|Average|
> |---|---|---|---|---|
> |w.o. Ms-PoE|52.6|53.2|64.0|56.6|
> |w. Ms-PoE|56.0|55.6|64.6|58.7|
>
>
> **[Q2: More Benchmark Results]** Thanks for the suggestions. We conducted additional experiments on all 13 tasks from the LongBench benchmark for further evaluation, and the results are presented in Table R2. Our findings show that Ms-PoE consistently demonstrated effectiveness across all 13 tasks, with improvements reaching up to 6.67 and an average of 2.2. It is important to note that we fixed $R_{min}=1.2$ and $R_{max}=1.8$ for all tasks without tuning on the test set.
>
> Table R2. Comparison results of LLaMA-2-7B-Chat with and without Ms-PoE on the LongBench benchmark. We use the same scaling ratios without further turning.
>
> | Methods    | MultiFieldQA-en | LCC   | GovReport | HotpotQA | Passage Count | Qasper | MultiNews | SAMSum | TriviaQA | PassageRetrieval-en | RepoBench-P | TREC  | 2WikiMQA | Average |
> | ---------- | --------------- | ----- | --------- | -------- | ------------- | ------ | --------- | ------ | -------- | ------------------- | ----------- | ----- | -------- | -------- |
> | w.o Ms-PoE | 33.51           | 59.77 | 27.97     | 30.10    | 3.74          | 19.27  | 24.36     | 39.45  | 82.81    | 10.00               | 49.22       | 57.33 | 28.14    | 35.82 |
> | w. Ms-PoE  | 37.33           | 62.03 | 29.87     | 34.08    | 4.60          | 20.96  | 24.69     | 39.79  | 85.28    | 16.67               | 50.11       | 58.67 | 30.19    | 38.02 |
>
>
> **[Q3: Determination of $R_{min}$ and $R_{max}$]** Thanks for the question, we determine the $R_{min}$ and $R_{max}$ via an ablation study on the MDQA task. Specifically,  we randomly selected 500 samples from MDQA tasks as the validation set and examined the effect of different scaling ratios, the results are reported in Table 4 where $R_{min}=1.2$ and $R_{max}=1.8$ demonstrates superior performance.  We then apply the same ratios to other downstream tasks without further adjustment.
>
>
> **[Q5: Welch’s t-test of Table 3]** Good suggestion. We conducted a further statistical analysis of the results presented in Table R3. As illustrated in Table R3, the baseline method shows a significant difference in performance when critical documents are positioned in the middle versus the beginning of the inputs, as well as between the middle and the end. However, the difference between the beginning and the end is not significant (p-value > 0.05). In contrast, when applying our Ms-PoE method, there is no significant difference in performance between the beginning, middle, and end positions, and it also achieves better average performance. These findings further confirm the effectiveness of our approach, and we have included these results in the updated manuscript.
>
> Table R3. Welch’s t-test of the results when different ordering metrics are applied. The Null hypothesis states that there is no difference between the means of two results.
>
> |      p-value      | Begin v.s. Middle | Begin v.s. End | Middle v.s. End |
> | :----------------: | ------------------ | -------------- | --------------- |
> |      Baseline      | 0.033              | 0.792          | 0.016           |
> |       Random       | 0.002              | 0.740          | 0.001           |
> |     Sequential     | 0.055              | 0.519          | 0.202           |
> |      Entropy      | 0.194              | 0.877          | 0.145           |
> | Position-Awareness | 0.391              | 0.562          | 0.151           |

---

> > ### Author Response · Authors · 2024-08-12
> > **Response to Reviewer LVQT**
> >
> > Dear Reviewer LVQT,
> >
> > We sincerely thank you for your time and effort in reviewing our work. Your constructive and insightful feedback has been invaluable in helping us improve the quality of our manuscript. We have carefully addressed each of your comments and hope that we have successfully resolved all of your concerns. We are open to further discussion if you have any additional questions and look forward to your response.
> >
> > Best regards,
> >
> > Authors

---

> > > ### Author Response · Authors · 2024-08-13
> > > **We are keen to discuss further with you**
> > >
> > > Dear Reviewer LVQT,
> > >
> > > We sincerely appreciate your valuable time and constructive feedback. We have carefully addressed each of your concerns. As the discussion period deadline approaches, we would be grateful if you could inform us if there are any further questions. Thank you!
> > >
> > > Best,
> > >
> > > Authors

---

### Official Review · Reviewer_xsrg · 2024-07-11

**Soundness:** 3
**Presentation:** 3
**Contribution:** 2
**Rating:** 6
**Confidence:** 4

**Summary:**

This paper proposes a plug-and-play method named Ms-PoE to mitigate the lost-in-middle challenge of LLMs. Specifically, Ms-PoE leverages multi-scale position embeddings to enhance information awareness in different parts of the context. Without fine-tuning the model,  Ms-PoE achieves an average accuracy gain of up to 3.8 on the Zero-SCROLLS benchmark over the original LLMs.

**Strengths:**

1. MS-PoE is a plug-and-play, simple yet efficient method, which is training-free and can achieve good performance.

2. The observation of "position-aware" attention heads is insightful, and applying re-scaling ratio dynamically to each attention head is reasonable.

3. I strongly agree with the authors' opinion that "the persistent difficulty faced by most LLMs in identifying relevant information situated in the middle of the context has not been adequately tackled."

**Weaknesses:**

1. It seems that the performance improvement on MDAQ and ZeroSCROLLS benchmark is small compared to the "Self-Extend" method, which can be seen as a special type of "Multi-Scale Positional Encoding", i.e., "Single-Scale Positional Encoding".

2. Although this paper suggests that Ms-PoE is suitable for "Long Contexts," I believe it is suitable for any context situation. What exactly defines a "Long Context"? Is it a context length above 12K? Additionally, I have not come across any analysis of the impact of context length and how Ms-PoE can help the model understand the middle of long contexts. It might be beneficial to conduct experiments to determine whether Ms-PoE can enhance the model's understanding of the intermediate content of the context. This could involve analyzing different context lengths, such as inserting a specific length of context and evaluating the model's predictions.

**Questions:**

1. What's the performance of MS-PoE(with LLama-2-7B-Chat as the backbone model) on LongBench-EN benchmark ?
2. Why chose 1.2 and 1.8 for R_{min} and R_{max} in Equation 2 ?
3. See Weakness 2

**Limitations:**

See Question and Weakness above.

---

> ### Author Rebuttal · Authors · 2024-08-07
>
> We appreciate Reviewer xsrg for acknowledging our method is “efficient”, and the observation of attention heads is “insightful”. To address Reviewer xsrg’s concerns, we provide pointwise responses in the following.
>
>
> **[Q1: Limited Improvements]** We respectfully disagree that our performance improvement is limited compared to “Self-Extend.” As demonstrated in Table 2, “Self-Extend” achieves similar performance to direct positional interpolation, or “Single-Scale Positional Encoding,” with gains of 1.20 and 29.76 on the MDQA and KV Retrieval tasks, respectively. In contrast, our Multi-Scale Positional Encoding achieves significantly higher gains of 3.92 and 43.72 on the same tasks, respectively, clearly surpassing the improvements achieved by “Self-Extend.”
>
>
> **[Q2: Ablation Studies of Context Lengths]** That’s a good suggestion. We conducted additional evaluations of our method across varying input lengths by changing the number of documents from 3 to 15. In these experiments, the key document is consistently positioned in the middle of the input to assess the LLMs' ability to capture middle context information. As depicted in Table R1, when the input length is short, our approach shows negligible improvement compared to the Baseline, with a 0.2 accuracy gain. However, as we increase the number of documents, thereby lengthening the input, significant improvements are achieved, with gains up to 6.4 in accuracy.
>
> Table R1: Results of the effectiveness of our approach across different input lengths. The relevant key documents are located in the middle of the inputs and experiments are conducted with Vicuna-7B
> |Number-of-Docs|3|5|7|10|15|
> |---|---|---|---|---|---|
> |Baseline|69.2|63.8|62.8|57.4|52.8|
> |Ours|69.4|64.4|67.0|63.0|59.2|
> |Improvements|0.2|0.6|4.2|5.6|6.4|
>
>
>
> **[Q3: Results on LongBench-EN Benchmark]** Thanks for the question. We conduct additional experiments of all 13 tasks from the LongBench-EN benchmark. Results are reported in Table R3. We can observe that Ms-PoE achieves consistent performance improvement without any finetuning.
>
> Table R3. Comparison results of LLaMA-2-7B-Chat with and without Ms-PoE on the LongBench benchmark. We use the same scaling ratios without further turning.
>
> | Methods    | MultiFieldQA-en | LCC   | GovReport | HotpotQA | Passage Count | Qasper | MultiNews | SAMSum | TriviaQA | PassageRetrieval-en | RepoBench-P | TREC  | 2WikiMQA | Average |
> | ---------- | --------------- | ----- | --------- | -------- | ------------- | ------ | --------- | ------ | -------- | ------------------- | ----------- | ----- | -------- | -------- |
> | w.o Ms-PoE | 33.51           | 59.77 | 27.97     | 30.10    | 3.74          | 19.27  | 24.36     | 39.45  | 82.81    | 10.00               | 49.22       | 57.33 | 28.14    | 35.82 |
> | w. Ms-PoE  | 37.33           | 62.03 | 29.87     | 34.08    | 4.60          | 20.96  | 24.69     | 39.79  | 85.28    | 16.67               | 50.11       | 58.67 | 30.19    | 38.02 |
>
>
>
>
> **[Q4: Determination of $R_{min}$ and $R_{max}$]** Thanks for the question, we determine the $R_{min}$ and $R_{max}$ via an ablation study on the MDQA task. Specifically,  we randomly selected 500 samples from MDQA tasks as the validation set and examined the effect of different scaling ratios, the results are reported in Table 4. We then apply the same ratios to other downstream tasks without further adjustment.

---

> > ### Comment · Reviewer_xsrg · 2024-08-11
> > **Respond to authors**
> >
> > Thanks for your response.
> >
> > Although Ms-PoE is an efficient and plug-and-play method, it may require some parameter-selection processes and the hyper-parameter may vary for different situations.
> >
> > I want to understand the impact of R_min and R_max. Do these two values significantly affect the final results?

---

> > > ### Author Response · Authors · 2024-08-11
> > > **Respond to further questions**
> > >
> > > Thanks for the good question. We conducted an ablation study to examine the impact of $R_{min}$ and $R_{max}$ on the final results. The findings are presented in Table 4 and discussed in Section 4.3 A2. Our study indicates that there is a sweet point for scaling ratios that enhances performance ($R_{min} = 1.2$, $R_{max} = 1.8$). However, when the scaling ratios are either too small (e.g., 0.5) or too large (e.g., greater than 2), performance tends to degrade. Additionally, such sweet point shows good generalization, as verified by downstream tasks such as key-value retrieval, Longbench, and Zeroscrolls.

---

> > > > ### Comment · Reviewer_xsrg · 2024-08-11
> > > > **Re-Respond to Authors**
> > > >
> > > > Thanks for informing me about this point. I have raised the score from 5 to 6.

---

> > > > > ### Author Response · Authors · 2024-08-12
> > > > > **Thanks for the response**
> > > > >
> > > > > We sincerely appreciate all the constructive feedback and positive evaluations from reviewer xsrg. Thanks for your time and support!

---

### Official Review · Reviewer_qA6u · 2024-07-12

**Soundness:** 3
**Presentation:** 4
**Contribution:** 2
**Rating:** 5
**Confidence:** 4

**Summary:**

This paper addresses the 'lost-in-the-middle' issue in large language models (LLMs) by introducing Multi-scale Positional Encoding (Ms-PoE). This approach enhances LLMs' ability to handle relevant information in the middle of the context without fine-tuning or added overhead. Ms-PoE uses position index rescaling and distinct scaling ratios for different attention heads to maintain essential knowledge. Experiments show Ms-PoE improves LLM accuracy, achieving up to a 3.8% gain on the Zero-SCROLLS benchmark.

**Strengths:**

1.	This paper is well-written and very easy to follow.

2.	The analysis of the position embedding sensitivity of different attention heads is very interesting and intriguing.

**Weaknesses:**

1．	The improvement is actually limited, despite the method’s efficiency and ease of use.

2．	The heuristic of the scaling ratio allocation process is a little arbitrary. There could be better solutions than just using an arithmetic sequence to achieve that. It is not strictly proved that a larger S_P means that the model should have a less rescaling ratio and vice versa. There could be possibilities that different heads could use the same scaling ratio or the other way around.

3．	Moreover, there is no evidence that the rescaled position embedding could make the attention heads like the “Top in Figure 4” better at catching relevant information except for an empirical marginal task performance improvement and the comparison with other strategies in Section 4.3. An attention map of the attention heads after the rescaling could better strengthen this main claim of the paper. The authors could also partially show this by examining the S_P variation before and after the rescaling.

4．	In Section 3.1, directly adjusting the hyperparameter and seeing the results on the test set is not very appropriate since it means the test set information is leaked. The authors should analyze the scaling ratio on the validation set and see if it is consistent with that in the test set. Moreover, the selection of the hypermeters R_min and R_max is also probably acquired by the test set performance, reflected by the ablation experiments in Table 4.

**Questions:**

1.	L212-215: “With a small scaling ratio LLMs tend to focus more on the most recent part of the context while with a large ratio, LLMs favor the beginning part” Are there any supporting results for this claim?

2.	Did you choose the hyperparameters of R_min and R_max directly from their performance on the test set?

3.	I’m happy to raise the scores if all the concerns are well addressed.

---

> ### Author Rebuttal · Authors · 2024-08-07
>
> We thank Reviewer qA6u for acknowledging our work as “interesting and intriguing”. We provide pointwise responses in the following.
>
> **[Q1: Limited Improvements]** We respectfully disagree with the claim that our improvements are limited. Our method offers a plug-and-play solution to enhance current open-source LLMs without any fine-tuning. And the enhancements are consistent across multiple benchmarks, including a 3.8-point average improvement on ZeroSCROLLS (Table 1), a 3.92-point increase in Multi-Document QA tasks, and a 43.72-point gain in KV Retrieval tasks (Table 2). Additionally, as shown in Table R1 in the uploaded PDF, our evaluations on the LongBench benchmark further demonstrate a consistent improvement of up to 6.67 points. Therefore, we believe our improvements are both consistent and significant.
>
> **[Q2: Studies of Scaling Ratio Allocation]** Thank you for the insightful comments. In our experiments, we chose the heuristic linear (arithmetic sequence) rescaling ratio due to its simplicity and input-independent nature. We can also rescale each attention head using other strategies, such as directly via the position-aware score ($S_P$) while it introduces additional computational costs. Since when implementing a scaling ratio based on the position-aware score, each input would have distinct rescaling ratio values. This would necessitate recalculating the $\theta$ value in RoPE on the fly, leading to additional computational overhead. Additionally, we can also allow different heads to share the same scaling ratio.
>
> To further address Reviewer qA6u’s concerns, we explored multiple scaling ratio allocation strategies and reported the results in Table R2. For the exponential and cosine strategies, we aimed to examine whether non-linear allocation strategies perform well. For the stepwise solution, we allowed different attention heads to share the same scaling ratio, and the position-aware score provided evidence for rescaling directly based on its raw scores. We observed that the position-aware strategy achieves similar results to the linear strategy but requires extra computational overhead. Similarly, allowing different attention heads to share the same scaling ratio yields comparable results. Therefore, based on the superior performance and computational efficiency, we chose the linear assignment method.
>
> **Details**: We compare several strategies in our experiments, including (i) Exponential: $r_i = 1.8 - (1.8-1.2) \cdot 0.9^{i}$, where $r_i$ is the i-th scaling ratio and 0.9 is the multiplication factor for scaling ratio decay. (ii) Cosine: $r_i = 1.8 - (1.8-1.2)cos(\frac{i}{n_h-1}\phi)$, where $n_h$ is the number of attention heads. (iii) Stepwise: $r_i = 1.2 + \frac{1.8-1.2}{3} round (\frac{i*4}{n_h})$, (iv) Position-Aware score: $r = 1.2 + \frac{(1.8-1.2)\cdot(S_P-min(S_P))}{max(S_P)-min(S_P)}$, as well as the Linear strategy in our current implementation.
>
> Table R2. Results of different scaling ratio strategies on the MDQA task with Vicuna-7B.
>
> |Key-Document-Index|1|3|5|7|10|Average|
> |---|---|---|---|---|---|---|
> |Baseline|64.0|61.0|57.4|58.4|64.8|61.12|
> |Ms-PoE (Linear)|65.6|64.2|63.0|65.2|67.2|65.04|
> |Exponential|64.0|60.0|63.6 | 63.6 | 63.6 |62.96|
> |Cosine|66.4| 63.0| 63.2 | 65.6 |66.4 |64.92|
> |Stepwise| 62.6|63.6 |63.6 |67.6 |68.0 | 64.88|
> |Position-Aware score|65.2|64.0|63.6 | 65.6 | 66.3 |64.94|
>
>
> **[Q3: Visualization of Attention Pattern before and after Rescaling]** Thank you for the suggestion. We visualized the attention patterns of different attention heads and reported the variation of $S_P$ before and after rescaling. The results are demonstrated in Figures R1 in the uploaded PDF. We found that the rescaling step effectively enhances the context utilization of LLMs, making some non-"position-aware" heads focus on critical information. Quantitatively, $S_P$ consistently increases with an average improvement from 1.79% to 1.93%.
>
>
> **[Q4: Determination of $R_{min}$ and $R_{max}$]** Thanks for pointing it out. We’d like to clarify that we didn’t search the best hyperparameter of $R_{min}$ and $R_{max}$ on the test set. Instead, we randomly selected 500 samples from MDQA tasks as the validation set and did the ablation studies of different scaling ratios on this validation set. We found that selecting a ratio between 1.2 and 1.8 significantly boosts the performance. Then. we apply the same ratios to all other tasks without further adjustment.
>
> **[Q4: Scaling Ratio Affects the Favored Zone]** Thanks. We report the raw accuracy results corresponding to Figure 3 in Table R3 where we position the key documents at the beginning, middle, or end of the sequences. From Table R3, we can observe that “changing the scaling ratio also affects the favored zone of LLMs. With a small scaling ratio (e.g., 0.5), LLMs tend to focus more on the most recent part of the context, while with a large ratio (e.g., 2.5), LLMs favor the beginning part”. We have included this table in the revised version for a clearer understanding of the content discussed in Lines 212-215.
>
> Table R3: Accuracy results for the MDQA task when key documents are positioned at various locations within the sequences. Different rescaling ratios are applied, with all attention heads sharing the same rescaling ratio.
> |Llama-2-7B-Chat | Beginning | Middle| End|
> |---|---|---|---|
> |0.5|36.6|41.8|59.4|
> |1|52.6|53.2|64.0|
> |1.5|59.0|58.4|59.8|
> |2|59.4|59.4|59.8|
> |2.5|62.4|55.6|56.0|
>
> |Vicuna-7B | Beginning | Middle| End|
> |---|---|---|---|
> |0.5|56.0|51.0|68.0|
> |1|64.0|57.4|64.8|
> |1.5|65.2|60.0|64.0|
> |2|61.5|59.0|62.5|
> |2.5|59.5|57.5|57.0|

---

> > ### Comment · Reviewer_qA6u · 2024-08-11
> >
> > Thanks for the responses. They resolved most of my concerns and I've raised my scores accordingly. Hope you all the best!

---

> > > ### Author Response · Authors · 2024-08-12
> > > **Thanks for the responses**
> > >
> > > Thanks for the responses. We are glad that our responses resolved your concerns and we will include additional results in the updated draft. Also, many thanks for raising the score.

---

### Author Rebuttal · Authors · 2024-08-07

We thank Reviewer qA6u, xsrg, and LVQT for their constructive suggestions and valuable questions. Additional supplementary materials are provided in the PDF, including:
- The attention patterns before and after rescaling **[Reviewer qA6u]**
- More results on LongBench. **[Reviewer qA6u, xsrg, LVQT]**

---

### Decision · Program_Chairs · 2024-09-25

**Decision:**

Accept (poster)

**Comment:**

This paper proposes a plug-and-play solution for the the "lost-in-the-middle" problem in large language models (LLMs) by introducing Multi-scale Positional Encoding (Ms-PoE). Ms-PoE uses position index rescaling and distinct scaling ratios for different attention heads to maintain essential knowledge. Experiments show Ms-PoE improves LLM accuracy on the several benchmarks.

Reviewers agree that this paper addresses an important problem and praise the simplicity and effectiveness of the proposed approach and the fact that it does not require any training. However they point out several weaknesses, such as the modest improvement in some tasks, the lack of analysis of performance in terms of context length, the lack of justification for the scaling ratio allocation heuristic (via an arithmetic sequence), and the process of tuning hyperparameters. Most of these issues have been convincingly addressed by the authors during the rebuttal, where they also added several other experiments (attention patterns before and after rescaling, new experiments in LongBench, and additional evaluations of the proposed method across varying input lengths). I am leaning towards acceptance.